# MOOC Behavior Analysis and Academic Performance Prediction Based on Entropy

**DOI:** 10.3390/s21196629

**Published:** 2021-10-05

**Authors:** Xiaoliang Zhu, Yuanxin Ye, Liang Zhao, Chen Shen

**Affiliations:** National Engineering Laboratory for Educational Big Data, Central China Normal University, Wuhan 430079, China; zhuxl@ccnu.edu.cn (X.Z.); yeyuanxin@mails.ccnu.edu.cn (Y.Y.); shenchen@mail.ccnu.edu.cn (C.S.)

**Keywords:** MOOCs, data mining, academic performance prediction

## Abstract

In recent years, massive open online courses (MOOCs) have received widespread attention owing to their flexibility and free access, which has attracted millions of online learners to participate in courses. With the wide application of MOOCs in educational institutions, a large amount of learners’ log data exist in the MOOCs platform, and this lays a solid data foundation for exploring learners’ online learning behaviors. Using data mining techniques to process these log data and then analyze the relationship between learner behavior and academic performance has become a hot topic of research. Firstly, this paper summarizes the commonly used predictive models in the relevant research fields. Based on the behavior log data of learners participating in 12 courses in MOOCs, an entropy-based indicator quantifying behavior change trends is proposed, which explores the relationships between behavior change trends and learners’ academic performance. Next, we build a set of behavioral features, which further analyze the relationships between behaviors and academic performance. The results demonstrate that entropy has a certain correlation with the corresponding behavior, which can effectively represent the change trends of behavior. Finally, to verify the effectiveness and importance of the predictive features, we choose four benchmark models to predict learners’ academic performance and compare them with the previous relevant research results. The results show that the proposed feature selection-based model can effectively identify the key features and obtain good prediction performance. Furthermore, our prediction results are better than the related studies in the performance prediction based on the same Xuetang MOOC platform, which demonstrates that the combination of the selected learner-related features (behavioral features + behavior entropy) can lead to a much better prediction performance.

## 1. Introduction

In recent years, massive open online courses (MOOCs) have received widespread attention thanks to their flexibility and free access [1], which has attracted millions of online learners to participate in courses. Although MOOCs have obvious advantages over traditional education channels, they still face many challenges [2]. One of the main challenges faced by MOOCs is the low learner completion rate. Research shows that the completion rates of MOOCs vary from 0.7% to 52.1%, with a median value of 12.6% [3]. Therefore, finding ways to improve the completion rates of MOOC learners has become a major challenge. Scholars have conducted in-depth studies on learners’ behaviors in the MOOCs environment and analyzed their learning state in the process of online learning, in order to gain a better understanding of learners’ course engagement, learning style, and behavior patterns. As part of the learning process, many MOOCs include video lectures, reading materials, quizzes, assignments, exams, and forum discussions [4]. Thus, the analysis of learners’ activities in these areas can reflect their behavior and active participation in the course [5], providing appropriate suggestions for improving learner engagement. Learner performance prediction has received much attention in recent years because the predictive learning outcomes of online learners can offer useful information for instructors to take timely interventions to get them through the course to completion [6].

A large amount of learners’ behavior data collected by some MOOC platforms have aroused scholars’ interest in data mining methods; therefore, data mining techniques are widely used in online learning behavior modeling to discover the relationship between learning behavior and learning outcomes in the learning process. Elbadrawy et al. [7] investigated two classes of methods for building prediction models, based on the behavioral data of students in MOOCs such as quizzes, assignments, and viewing video lectures, which can predict student performance in a timely and accurate manner. Chamizo-Gonzalez et al. [8] analyzed the online learning behavior data of 129 students in Moodle and found that behaviors of actively uploading assignments, publishing learning resources, and actively posting on the course forums can improve students’ learning outcomes. Al-Musharraf et al. [9] applied the naïve Bayes classifier on learners’ records and found that learners’ behaviors, including attending live virtual lectures and taking online quizzes, are positively correlated with their academic performance. Thus, we can infer that learning behavior is an important representation of the learning process that can accurately describe the learning state of learners and affect their academic performance. However, the existing research lacks an in-depth discussion on the relationship between the changing trend of learning behaviors of learners and academic performance in MOOCs.

In this study, we analyze the online learning process of learners and extract behavioral features from their behavior records in order to describe the online learning states of learners as accurately as possible. Next, we use entropy to describe the change trends of the learners’ corresponding behavior in a certain activity and explore the relationships between entropy and behavioral features, as well as between features and academic performance. In addition, we use cluster analysis to describe the main behavior characteristics of learners, and compare them with behavior entropy to further illustrate the consistency between behavior entropy and representative behavior characteristics in describing group behavior participation, indicating the effectiveness of entropy in quantifying behavior trends.

We use an appropriate feature selection method to reduce the number of features by eliminating redundancy and conduct an experiment of academic performance prediction in a dataset of 12 MOOCs. The results show that the classification prediction model can identify the key features and achieve a good level of prediction performance. In order to verify the importance of the predictive features in the prediction process, we compare experimental results from relevant research. The results show that, even though the datasets used in the experiment are similar, our model has better predictive performance in all courses combined. It shows that behavioral features related to learners play an important role in improving the predictive performance of models. Our main contributions are summarized as follows:

(1) We use entropy to describe the change trends of learners’ behavior and to explore the relationships between entropy, behavioral features, and academic performance. In the cluster analysis experiment, we use a k-means algorithm to explore the main behavior characteristics of learners, use visual means to describe the data distribution of group behavior and entropy, and then compare and analyze the consistency between behavior and corresponding entropy.

(2) In the academic performance prediction, we use the appropriate feature selection method to reduce the number of features, removing redundancy, so that the prediction performance of the model is good.

(3) We conduct experiments to compare our results with other related research and demonstrate the importance of predictive features related to learners.

## 2. Related Work

In academic performance prediction in MOOCs, some researchers have examined the problem from the perspective of using predictive variables to explore the relationship between behavior and academic performance. Marbouti et al. [10] used seven different predictive modeling methods to identify at-risk students early, so that tutors can then take preventative action. The input features used in this study for performance prediction included grades for attendance, quizzes, weekly homework, team participation, project milestones, mathematical modeling activity tasks, and exams. They found that the naïve Bayes classifier model and an ensemble model using a sequence of models had the best results among the seven modeling methods tested. Conijn et al. [11] analyzed the relevant literature and summarized predictor variables for performance prediction. According to log data in the Moodle platform, they extracted predictive variables such as page viewing, resource viewing, quiz, assignment, wiki, and forum discussion to predict final exam grades using a multiple linear regression model. Minh-Duc et al. [12] forecasted students’ learning outcomes using four regression models. They found that four factors, the number of views, the number of posts, the number of forum views, and the number of on-time submitted assignments, impacted the students’ learning outcomes.

When describing learning behavior, it is also necessary to describe the changing trend of learning behavior in order to explore the behavior uncertainty or disorder of learners in the learning process in MOOCs. As a statistic in the field of information theory that effectively quantifies the overall trends of a time series, entropy [13] reflects the uncertainty of the occurrence of new information in the series. In the field of education research, researchers have used entropy analysis to reveal the educational trends of learners under different learning scenarios. Cao et al. [14] collected the campus smart card records of 18,960 undergraduate students and applied entropy to measure the orderliness of college students’ daily life and learning activities on campus. The research results showed that the orderliness of behavior quantified by actual entropy is an important feature in predicting academic performance. Zhang et al. [15] used the hidden Markov model to model the behavior sequence of learners, and the hidden states represented the latent behavior regularity. They used entropy of transition to represent the transition from a certain behavior to another, which reflects the hidden regularity of daily life and habits. San Pedro et al. [16] used entropy to quantify the fluctuating trend of interaction patterns such as emotion, behavior, knowledge, and correctness of students in the math tutoring system. Research showed that these interaction patterns quantified by entropy analysis are significantly correlated with students’ final exam scores. In this study, in order to better quantify the change trend of learners’ behavior in MOOCs, we use entropy as the main measurement to represent the uncertainty of the distribution of various activities represented by a certain behavior.

## 3. Methods

### 3.1. Data Description

Data about students’ online behavior were collected from the Xuetang platform, the largest MOOC platform in China, in the four quarters (spring and autumn) of cohort 2017–2018. From this period, 12 courses were selected as the research objects. We initially obtained the behavior data of 76,843 learners by collecting and processing the server log data and recording information of the corresponding courses. There are, however, a large number of registrants in the MOOC learning process [17] who register but do not undertake any courses, and thus are without any learning records. Therefore, in order not to affect the accuracy of the experimental results, all learning behavioral records of the registrants need to be deleted. These courses mainly belong to the fields of mathematics, English, and computers. The main learning forms of courses are divided into lecture viewing and test submission. The specific information of these courses is shown in Table 1.

### 3.2. Behavioral Feature Extraction

#### 3.2.1. Statistical Behavioral Features

It is necessary to depict the main behavior patterns of learners’ participation in MOOC activities by the statistical behavioral features. Thus, we can obtain learners’ behavioral features through the descriptive statistical results of behavioral data, such as frequency, average, variance, and so on, which represent learners’ general learning rules. According to the data collected on the MOOC platform, the course activities that learners participate in include viewing course videos, submitting quizzes, participating in discussion forums, searching web pages, and logging into their account. Thus, in this study, we designed 19 descriptive statistical behavioral features based on the MOOC dataset to characterize the learning status of learners, as shown in Table 2.

It is relatively intuitive to evaluate the behavior by the statistical features of the main behaviors of the learners participating in MOOC, including the overall status, average level, and regularity. In terms of the overall situation of behavioral features, such features should reflect the final overall status of learners in a certain behavior. For instance, progress of quizzes submitted (QP) is a measure of the final progress of learners after submitting quizzes, which is obtained by the proportion of the number of submitted quizzes to the total number of quizzes in the course. Similarly, the progress of learners watched different videos in a course (VP) is mainly obtained by calculating the completion rate of learners watching video lectures during the course, which is calculated by dividing the number of videos that have been watched completely by the total number of videos that exist in the course. The average time of video views (VT) represents the average time duration that learners watch video lectures each time during the learning process. Rate of video views (VR) represents the proportion of the time recorded by the platform to the current learning time in the process of learners watching video lectures. If a learner watches the course for less time than the total time of the course video he/she is watching and the position from which he/she is watching is not the starting position, according to the condition, we count the total number of videos in which the learner dragged the video progress bar. Probability of video playback progress bar moved (VMP) is the proportion of the total number of videos in which learners dragged the progress bar to the number of videos he/she has watched. In addition, there are some statistical data of some course activities completed by learners on the learning platform, such as the number of forum posts (PN), the number of forum replies (RN), the number of content searches (SN), and so on. Such features are usually obtained by recording the sum of the number of times learners participate in activities.

In terms of the average level of behavioral features, such features represent the average level of learners’ participation in course activities in a period of time. For example, score rate of quizzes submitted (QSR) is calculated by the ratio of the scores evaluated by the instructors to the total scores after the learners submit the quizzes, which reflects the average accuracy of the learner completing the test many times. Average time interval of video views (VAT) is the average time interval for learners to watch video lectures during the course of studying. Obviously, for more active learners, the time interval of watching lectures is shorter. A similar feature is average time interval of quizzes submitted (QAT), which represents the average time interval between multiple submissions of quizzes by learners. Because there is a certain correlation between the behavior of watching a video and the behavior of submitting quizzes, similar behavioral features are no longer selected. From another perspective, max time interval of quizzes not submitted (NQT) represents the longest time interval for learners to submit quizzes multiple times in a period of time. In general, for inactive learners, the interval for not submitting quizzes is longer than for active learners. Similarly, time interval until first video views (VFT) is obtained by the time interval from the time a learner chooses a course to the first time he/she watches the video lecture. Overall, such features often reflect the learners’ enthusiasm in learning the course. When the learners’ enthusiasm for participating in the course activities is higher, the feature value is smaller; in the opposite case, the feature value is larger.

In terms of the regularity of behavioral features, such features reflect the regularity of learners’ participation in course activities in a period of time. For example, variance of time interval of video views (VTV) represents the variance of the time interval for a learner to watch video lectures several times during the course. Because learners may learn courses at any time, the feature can reflect the regularity of learners in learning courses to a certain extent. The smaller variance value indicates less volatility, and thus stronger regularity.

#### 3.2.2. Temporal Behavioral Features

Regarding the learners’ temporal behavioral data, nonlinear metrics are used to examine these behavioral patterns. In this study, entropy is used as the main method of research in order to better quantify the change trends of learners’ behavior in MOOCs. To some extent, entropy indicates that learners are in an “anytime” learning state, which means that learners may participate in the corresponding course activities. We can get the specific entropy of a certain behavior of learners participating in MOOCs activities. The equation of entropy is defined as follows [16]:(1)$$\mathrm{Entropy}\left(X\right)=-\overset{N}{\underset{i=1}{\sum }}p\left(x_{i}\right)log_{e}p\left(x_{i}\right)$$
(2)$$p\left(x_{i}\right)=\frac{Count\left(x_{i}\right)}{N}$$
where $p\left(x_{i}\right)$ represents a certain stated probability of learners’ course learning behavior, *N*represents all state sets, and $Count\left(x_{i}\right)$ represents the number of repetitions of $x_{i}$. By analyzing the course design and assessment methods of these 12 courses, we find that the most important course activities of learners include watching videos, submitting quizzes, and interacting in forums. Thus, we obtain the corresponding behavior entropy according to the three main behaviors.

In the analysis of behavior entropy, we firstly focus on the record of a learner’s behavior throughout the day (i.e., {t_1_, t_2_, ···, t_n_}). Subsequently, we divide one day into 48 time bins such that each bin spans 30 min [14,18,19]; therefore, every bin is encoded from 1 to 48 (i.e., t_i_′∈ {1, 2, ···, 48}, where i denotes the i^th^ time bin). Then, the time series {t_1_, t_2_, ···, t_n_} can be mapped into a discrete time sequence {t_1_′, t_2_′, ···, t_n_′}. Next, we can get the entropy E_1_. The entropy of video views that the learner accessed in course lectures in a semester is {E_1_, E_2_, ···, E_T_}, where T denotes the number of days when the learner viewed course videos. Let us begin by considering a simple case in which the time series of a learner’s course videos viewing in one day is {7:30, 7:42, 9:38, 10:31, 15:43, 20:25, 22:43, 23:12}, i.e., the learner participated in the learning activity in the following eight different time slots: {15, 16, 20, 22, 32, 41, 46, 47}; then, we can calculate the probability value by *p*(*x_i_*) = 1/8, i∈ [1,8]. However, in the case that the learner participated in more than one learning activity during a certain time slot, the probability value is different from the previous case. For example, in the behavior sequence {15, 15, 20, 22, 32, 41, 46, 47}, *p*(*x_i_* = 15) = 2/8. Obviously, this is similar to the behavior trend of learners’ daily participation in course activities; that is, when learners are keen to participate in a certain learning activity, more records can be captured during the same learning time slot. The greater the probability of occurrence, the smaller the entropy (entropy is inversely proportional to probability). Notably, small entropy indicates that learners are likely to participate in learning activities at a specific time regularly; At the same time, according to the probability value of the above two cases, the behavior entropy of the learner can be calculated respectively according to the definition formula of entropy. However, in the MOOC scenario, only few learners are able to participate in regular learning, because not all learners can learn at a specific time regularly (affected by time, work, life and other factors), so we assume that learners are more likely to learn at any time, without any regularity in a certain day. This assumption could be verified by the behavior entropy of learners with large entropy values. Therefore, considering that the time of participating in activities is random and the distribution of time characteristics is neither concentrated nor regular, we use information entropy to describe the change trend of learners’ behavior in this study.

## 4. Experiments and Results Analysis

### 4.1. Correlation Analysis

Learners’ behavior patterns reflect their learning states to some degree and have an important impact on their performance [15]. In this section, we describe the relationships between learner’s behavior, entropy, and academic performance through correlation analysis of two aspects. Firstly, in the correlation analysis of learners’ statistical behavioral features and corresponding entropy, we attempt to obtain an internal correlation between them. When learners participate in different course activities, statistical behavior features obtained from behavioral data can reflect learners’ course participation and learning styles [20]. If the behaviors described by statistical features are closely related to entropy, both entropy and behaviors can reflect course engagement to a certain extent. Secondly, it shows the relationships between learners’ behaviors and academic performance in different courses through the correlation analysis between behavioral features and academic performance, and explores which behaviors are closely related to academic performance.

#### 4.1.1. Between Entropy and Statistical Behavioral Features

In the aspect of related learning behavior, we use entropy to measure the learner behavior disorder or the characterization of learners in MOOC randomness/disorder in the process of learning. As the behaviors of MOOC learners are affected by themselves, courses, environment, and so on, they show a certain regularity in terms of behaviors; therefore, there may be correlations between these behaviors. In order to explore whether there is an internal correlation between these features and the entropy of corresponding behaviors, we conduct a correlation analysis on them. The correlation coefficient reflects the direction and degree of the change trend between the two variables. The larger the absolute value, the stronger the correlation. As there may be a nonlinear relationship between behavioral features, we use the Spearman rank correlation coefficient [21] to measure the degree of correlation between features. In order to facilitate the understanding of the relationship between features and entropy, the experimental results are displayed using the correlation coefficient heat map.

Figure 1a describes the correlation coefficient between entropy and its corresponding statistical behavioral features. We find that the number of forum posts (PN) and the number of forum replies (RN) are highly correlated with the number of words posted in forum (PWN) and the number of words replied in forum (RWN) (r > 0.95). The behavior entropy of forum interaction (FE) has a certain correlation with each behavioral feature (r > 0.2), which has the highest correlation with the number of forum replies (FN). As shown in Figure 1b, the behavior entropy of quizzes submitted (QE) is correlated with other quiz-related behavioral features (r > 0.39), which has the highest correlation coefficient with the average time interval of quizzes submitted (QAT). However, there is a high correlation between the max time interval of quizzes not submitted (NQT) and the average time interval of quizzes submitted (QAT) (r > 0.8).

As shown in Figure 2, the behavior entropy of video views (VE) has a certain correlation with related video viewing behavioral features, except for the weak correlation with multiples of video playbacks (VM), the variance of time interval of video views (VTV), and the number of repeated video views (VRN). For example, it is highly correlated with the progress of video views (VP), rate of video views (VR), and average time of video views (VT), and negatively correlated with the time interval until first video view (VFT) and the probability of video playback progress bar moved (VMP). Overall, the results show that there is a certain correlation between each behavioral feature and the corresponding behavior entropy, which indicates that the behavior entropy and behavior are related and some behavioral features are highly correlated. Thus, temporal behavioral features represented by entropy can measure the behavioral change trends of learners in some aspects and reflect the course engagement of learners as well as some key behavioral features.

#### 4.1.2. Between Academic Performance and Behavioral Features

It is necessary to delete some highly correlated features to reduce the impact on the experimental results because of redundancy or coupling phenomenon in each behavioral feature. According to the above analysis results, we should delete the highly correlated features such as the number of words posted in forum (PWN), the number of words replied in forum (RWN), the average time interval of quizzes submitted (QAT), the average time of video views (VT), and the number of video views (VN). Then, we carry out Pearson correlation analysis on the selected features and the academic performance of course combination and each course, in order to explore the relationships between the features of each course and course combination on the academic performance. The experimental results are shown in Table 3.

Table 3 describes the Pearson correlation coefficient between academic performance and the behavioral features for all courses combined. The experimental results show that the number of forum posts (PN), the number of forum replies (RN), and the average time interval of video views (VAT) have no significant impact on the academic performance between the course combination and each course. There is a negative correlation between academic performance and features such as the number of context searches (SN), the number of repeated video views (VRN), the probability of video playback progress bar moved (VMP), the variance of time interval of video views (VTV), and the time interval until first video view (VFT) at a certain significant level (r < 0, *p* < 0.001), while other features are positively correlated with academic performance for all courses combined (r > 0, *p* < 0.001). Among these features, the progress of quizzes submitted (QP), the score rate of quizzes submitted (QSR), the behavior entropy of quizzes submitted (QE), the rate of video views (VR), and the behavior entropy of video views (VE) have a strong positive correlation with academic performance (r > 0.6). Progress of video views (VP) and average time of video views (VT) are also correlated with academic performance. In terms of learning behavior, the behavioral features related to course video viewing and quizzes submitted are correlated with academic performance at a significant level, which indicates that learners pay attention to behaviors such as chapter tests and course lectures in different courses. Behavioral features related to forum interaction, such as the number of forum posts (PN) and the number of forum replies (RN), do not correlate with academic performance, which indicates that the participation of learners in the forum is generally low and often confined to a minority of learners. However, even with more active engagement strategies to encourage participation, it seems that active forum behavior appeals to some learners, but not to others [22]. In terms of behavioral features, time interval until first video view (VFT) and variance of time interval of video views (VTV) are negatively correlated with academic performance, which indicates that the learning enthusiasm and regularity represented by the features have an impact on course learning when the learners viewed course videos. However, the max time interval of quizzes not submitted (NQT) shows a certain correlation with academic performance (r > 0.3), indicating the learning inactivity measured by the feature can also affect the learner’s course learning. In general, the results show that it may be useful to compare students not only with their peers, but also with their own behavior in other courses [11]. According to the statistics of course information, 1011 learners took two courses C4 and C10 at the same time, from which it can be explored whether there is a difference in the relevance of the same behaviors to academic performance in the two courses. The data in Table 3 show that, when learners took the C4 and C10 courses together, the difference between the same behavior and academic performance is mainly reflected in forum interaction and course video viewing. For example, in the C4 course, there is a significant correlation between academic performance and behavioral features such as forum interaction and course video viewing (*p* < 0.01). However, in the C10 course, there is no significant correlation between academic performance and the partial behavioral features such as forum interaction and course video viewing, which indicates that the overall learning status of learners may be affected by learning styles [23] and course design [24].

In summary, the indicator of behavioral change trends measured by entropy can effectively characterize the states of learners (i.e., randomness or disorder). The results show that it has a certain correlation with academic performance. To some extent, each behavioral feature can also represent the states of learners, such as learning enthusiasm, regularity, and irregularity, and these states may have a certain correlation with academic performance. This indicates that, in the process of online learning, the learning states represented by behaviors can measure a certain learning stage of the learner.

### 4.2. Cluster Analysis

To further verify the effectiveness of entropy in quantifying behavioral trends, we will use a clustering algorithm to conduct an analysis on learners’ behavioral data. Then, to further verify the consistency between the behavior entropy features and the representative behavioral features, which represent group behaviors from different perspectives, the corresponding comparison is performed.

#### 4.2.1. Clustering Algorithm and Indicator

As one of the essential data mining methods, cluster analysis aims to divide a data set (represented by multiple features) into different clusters so that the objects in the same cluster are more similar to each other compared with those in other clusters [20]. In general, cluster analysis mainly consists of three core parts. First, some appropriate cluster indicators are selected, representing learners’ behavioral characteristics. Then, a more stable clustering algorithm is applied, which can reasonably divide learners into different groups. Finally, the differences of behavioral characteristics among different learner groups are explained by analyzing the clustering results.

According to the correlation analysis in Section 4.1, three representative statistical features are selected to represent the main course behaviors of learners initially. Further, we convert the number of video views (VN), the number of forum posts (PN), and the number of forum replies (RN) into the progress of video views (VP) and the number of forum interaction (FN), in order to distinguish the course commitment of learners. Thus, the final indicators used for clustering are VP, FN, and the progress of quizzes submitted (QP). At the same time, in order to eliminate the impact of magnitude data, the data are standardized using the zero-mean normalization method. Finally, the K-means algorithm is chosen as the clustering algorithm in our experiment, thanks to its stability, reliability, and wide usage.

#### 4.2.2. Selecting the Optimal Number of Clusters

The main problem faced during the process of cluster analysis is to estimate the correct number of clusters by appropriate methods. One of the proper solutions to solve this problem is the elbow point method [25], which calculates the cost function as follows:(3)$$\mathrm{Cos}t=\log\left[\frac{1}{n}\overset{k}{\underset{i=1}{\sum }}\underset{x\in Cluster_{i}}{\sum }{\left|x-Centroid_{i}\right|}^{2}\right]$$
i.e., taking the sum of square distance errors (i.e., average distortion) between the particle of each cluster and the sample points in the cluster.

Obviously, given a cluster, the lower the cost (i.e., average distortion), the closer the structure (between members in the cluster) becomes; conversely, the higher the cost function, the looser the structure becomes. Generally, the cost will decrease with the increase in the number of clusters. However, given data with a certain degree of differentiation, the total distortion will first be greatly improved when it reaches a certain critical point, and then it will slowly decline, i.e., the critical point can result in optimal clustering performance. Meanwhile, in order to verify the effectiveness of the correct number of clusters in the K-means algorithm, the silhouette coefficient [26] is adopted as an important index combining the cohesion and separation as follows:(4)$$S\left(i\right)=\frac{b\left(i\right)-a\left(i\right)}{\max\left\{a\left(i\right),\;b\left(i\right)\right\}}$$
where *a*(*i*) is the average distance between the *i*th sample point and other sample points in the cluster. The smaller *a*(*i*) shows the *i*th sample point is more likely to be clustered to the current cluster. *b*(*i*) is the average distance between the *i*th sample point and all the sample points in other clusters. The larger *b*(*i*) indicates that the *i*th sample is less likely belong to other clusters. In the case when the value of silhouette coefficient (ranging from −1 to 1) is close to 1, this indicates that a good overall quality of the clustering algorithm is obtained. That is to say, the sample points in the cluster are close to each other, but far away from other clusters.

Figure 3 describes the selection of cluster number based on both the elbow method and iterative evaluation of clustering performance. Obviously, from Figure 3a, we can find that the inflection point at the elbow is 4, which indicates that the optimal number of clusters is 4. In addition, the validation of clustering performance in Figure 3b shows that the clustering number of 4 or 5 can lead to higher clustering performance. Thus, 4 was finally chosen as the number of clusters in the experiment.

#### 4.2.3. Clustering Results

The K-means algorithm along with the above clustering indicator (i.e., QP, VP, FN) is used to divide the learners into four groups, namely, O1, O2, O3, and O4. Table 4 shows the centroid values of the four groups obtained. In order to better understand the specific behavioral characteristics of the four groups, the data distribution diagram of each indicator in the four groups is constructed, as shown in Figure 4a.

In Figure 4a, compared with learners in other groups, learners in group O1 have the lowest engagement in course activities such as submitting quizzes, watching videos, and interacting with forums, which indicates that Q1 is a common low active group in MOOCs. As shown in Table 4, VP in group O2 is between those in groups O3 and O4, but FN in group O2 is much higher than those in the other three groups. This indicates that the learners in group O2 are more inclined to interact in the forum, and they like to express their opinions in the forum, while the engagement in other course activities is at the middle level. Most learners in group O3 completed the quizzes at a higher rate and its QP is the second largest—only lower than that in group O4. In contrast, VP in group O3 is lower than those in groups O2 and O4, and FN is slightly higher than that in group O4, which actually shows a group lack of community interaction. The behavior characteristics of group O3 show that learners in this group are good at completing both the course assignments and quizzes assigned by instructors, but they are not interested in watching videos and interacting with the forum. Additionally, learners in group O4 are very active in completing quizzes and watching videos, i.e., their QP and VP are higher than those in other groups, while FN is lower than that in groups O2 and O3. These results show that learners in group O4 are more inclined to acquire knowledge through traditional learning methods such as watching videos, completing homework, and quizzes, and are not keen on communication and discussion in the online community, which is a typical performance-oriented learning pattern.

In order to further verify whether there is a consistent relationship between learners’ MOOCs behavior and its corresponding entropy, we conduct visual analysis of learner groups’ clustering data distribution based on the behavior entropy. As shown in Figure 4b, the distribution of behavior entropy among each learner group is basically consistent with the distribution of indicator data in Figure 4a. Taking quizzes submission behavior as an example, QP in groups O3 and O4 are higher, followed by group O2 and group O1 (Q1 has the lowest completion progress). Obviously, from Figure 4b, we can see that, firstly, the group behavior entropies of groups O3 and O4 are in a higher range, while those of group O2 and O1 are in a lower range. At the same time, the entropy distribution of other group behavior also shows similar characteristics. Secondly, the entropy distribution is not concentrated in a small range, i.e., more well-distributed than those of the three indicators shown in Figure 4a. This situation indicates that, compared with the clustering indicators, the behavior entropy of learners is more suitable to represent the time distribution of learners’ course activities. This is because most MOOCs learners may participate in course activities at any time, which is reflected in the random and decentralized distribution of entropy in terms of time.

### 4.3. Predicting Academic Performance

As it is proved that behavior entropy is not only correlated to academic achievement, but also has the ability to quantify group characteristics, it can be used to predict academic achievement combined with basic behavior characteristics. In this section, we take the final grades of learners as the evaluation indicator of their academic performance and take whether or not a pass grade was achieved as the binary classification prediction standard. Accurate modeling can help to predict whether learners will drop out of the course in high-risk situations, and timely and effective teaching intervention means can be adopted to reduce the rate of dropout for learners.

#### 4.3.1. Feature Selection

In the modeling process of machine learning algorithms, feature selection is a process of selecting a subset of relevant features [27], because unnecessary features will affect the generalization ability of the model and increase its computational cost. In order to maximize the prediction performance, domain knowledge should be provided as support to the allocation of the best performing sets of input data or learner-related features [28,29]. We use the correlation coefficient method to select the features for classification prediction. Firstly, we delete the feature that the absolute value of Pearson correlation coefficient is less than 0.2, which means that the correlation with academic performance is weak. We find that the correlation coefficient between progress of quizzes submitted (QP) and academic performance is 0.90, while the grades of MOOC learners were finally obtained by the sum of the grades of test assignments according to a certain proportion of weight. Therefore, this phenomenon of high correlation indicates that learners will have good grades as long as they submit quizzes, which may be related to the simplicity of the course assessment form. Therefore, the progress of quizzes submitted (QP) should be deleted in order to reduce the risk of overfitting. Finally, the features that can be used for classification prediction of all courses combined are the score rate of quizzes submitted (QSR), the entropy of quizzes submitted (QE), the max time interval of quizzes not submitted (NQT), the progress of video views (VP), average time of video views (VT), the rate of video views (VR), the variance of time interval of video views (VTV), the time interval until first video view (VFT), the entropy of video views (VE), and the entropy of discussing in forum (FE). Similarly, we used the same strategy to select features with higher correlation coefficients as input variables for classification prediction of different courses.

#### 4.3.2. Baselines

To verify the importance of different feature combinations related to learners in classification and prediction, we compare the prediction performance with those of previous related research based on the same Xuetang MOOC platform [30]. Similar to [30], on the one hand, we chose four commonly used prediction methods as the benchmarks to predict learner performance of 12 courses, including logistic regression (LR) [31], support vector machine (SVM) [32], decision tree (DT) [33], and random forest (RF) [34]; on the other hand, we used a combination of the features selected (behavioral features + behavior entropy) as the input variables for predictive analysis. Although the data sets used are different and the experimental methods designed are also different, they still belong to the course data of the same platform. We assume that they own similar data distribution laws. Therefore, the results still have reference value.

#### 4.3.3. Metrics

In this study, we use precision, recall, F1 score, and AUC as evaluation metrics for binary classification prediction. Precision represents the proportion of samples that are positive among all the samples whose predictions are positive; recall indicates the proportion of samples that are predicted to be positive among all samples that are actually positive; and the F1 score represents the harmonic mean of precision and recall, which measures the predictive performance of the classification algorithm. The formulas of precision, recall, and F1 score are defined as follows:(5)$$Precision=\frac{TruePositive}{TruePositive+FalsePositive}$$
(6)$$Recall=\frac{TruePositive}{TruePositive+FalseNegative}$$
(7)$$F1\;\mathrm{score}=\frac{2*Precision*Recall}{Precision+Recall}$$

In response to the data imbalance in the dataset, the measured metrics are a poor measure of performance such as precision or recall; thus, we use AUC as the comparison standard for different algorithms.

#### 4.3.4. Experiment Details

In the research process, it is necessary to preprocess the data to improve their quality. The main processing stages are outlier processing, missing value processing, and data normalization. We analyze the causes of data quality variance and use reasonable methods to deal with outliers. At the same time, there are missing values in the data, which means that there is no corresponding behavior record. Thus, it is a feasible method to fill in the missing values with a zero value. As the different features were initially at different scales, in order to eliminate the influence of scales between features, we use the Z-score standardized method to normalize the data to ensure that the data are on the same scale. The equation of Z-score standardization is defined as follows:(8)$$x^{\prime }=\frac{x-mean}{std}$$

In the process of parameter selection, we use fivefold cross-validation based on a grid search to select the optimal parameters. The coefficient C of LR with l2 penalty is 0.1. The penalty coefficient C of SVC with RBF kernel is 100 and the kernel function parameter γ is 0.01. The parameter n_estimators of RF with entropy criterion is 50 and the maximum depth max_depth of the DT is 6. In the process of classification prediction, we randomly divide each data set into a training set (75%) and a test set (25%). Then, we train the models with optimal parameters obtaining by a grid search and use fivefold cross-validation to evaluate the predictive performance of the model. Finally, the implementation and comparison of the different models we use are supported by the machine learning library scikit-learn [35].

#### 4.3.5. Results

Table 5 describes the results of the comparison of academic performance prediction and evaluation indicators of each model in the course combination and different courses. 

Table 5 shows, compared with the baseline, that a better effectiveness in predicting learners’ curriculum achievement can be obtained using our proposed method, which indicates that different feature combinations related to learners play an important role in prediction. We can also see that, generally, the prediction results with entropy features (i.e., the fifth row in Table 5) outperform the results without entropy features (i.e., the fourth row in Table 5). In additional, better combinations can also improve the prediction performance. Therefore, by using various behavior indicators of learners, we can quantify the behavior state of learners and forecast the corresponding learning outcomes.

Furthermore, among the four classifiers shown in Table 5, RF outperforms the other three classifiers in all predicted performance indexes; the performance of SVM, DT, and LR in predicting performance decreases in turn, but the gap is not very large. In general, each model has a good performance in the classification prediction of academic performance, which indicates that these models can identify the key behavioral features of learners in the prediction process. Secondly, according to the results in Table 5, although the prediction performance of each model in most of the courses is good, the prediction performance in the C11 course is weaker, which may be due to the lack of sufficient data and key features related to learners in this course. Therefore, the above results show the effectiveness of features in predicting academic performance. We infer that the behavior of learners can reflect their learning states in MOOCs and help predict the performance of learners to prevent dropout.

## 5. Conclusions

In this study, we first analyze the relevant literature on learners’ behavior features and academic performance prediction in MOOCs. Next, we propose a quantitative behavioral change trends indicator behavior entropy based on existing datasets. We then conduct correlation analysis between the behavioral features, the corresponding entropy, and academic performance. The results show that there is a correlation between entropy and the corresponding behavioral features; thus, the change trends of behavior quantified by entropy can represent the behavioral states of learners to some degree. At the same time, in order to further explain that entropy can describe the behavior change trend, we use cluster analysis to explore whether there is consistency between learners’ group behavior and entropy in terms of data distribution, that is, whether entropy can further reflect learners’ behavior participation. Finally, in order to verify the effectiveness of features related to learners in performance prediction, we use four models to predict academic performance, and the results show that these models can identify the key behavioral features of learners in the prediction process. In the comparison of the prediction results of different courses, we conclude with the effectiveness of features in predicting performance.

Although the role of behavior entropy in academic prediction has been confirmed in this study, owing to the possible differences in data attributes caused by, for example, privacy protection across diverse data platforms (e.g., other platforms rather than Xuetang MOOC used in this study), our proposed model has limitations in further promotion.

## Figures and Tables

**Figure 1 sensors-21-06629-f001:**
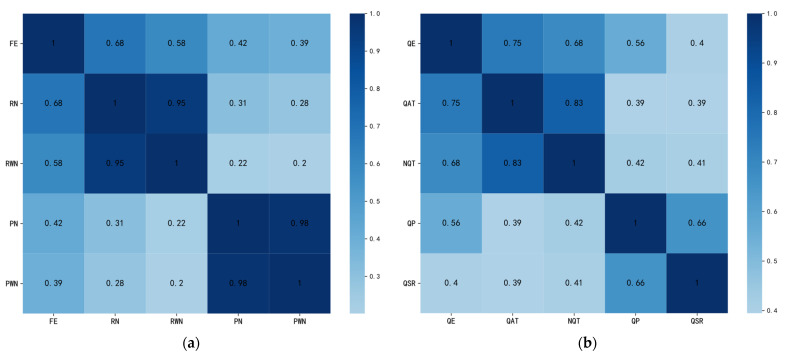
Heat map of two main behaviors: (**a**) forum interaction and (**b**) quizzes submission.

**Figure 2 sensors-21-06629-f002:**
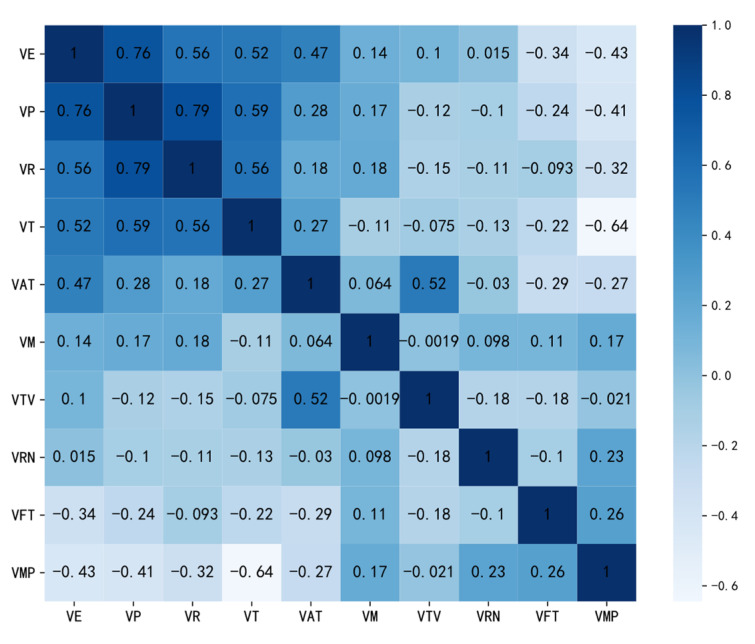
Heat map of lecture video viewing behavior.

**Figure 3 sensors-21-06629-f003:**
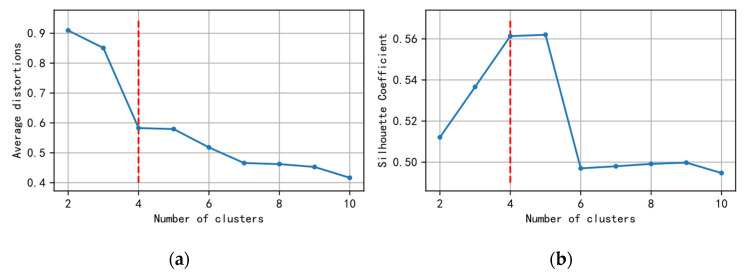
(**a**) Selection of cluster number with the elbow method and (**b**) validation of clustering performance using the silhouette coefficient.

**Figure 4 sensors-21-06629-f004:**
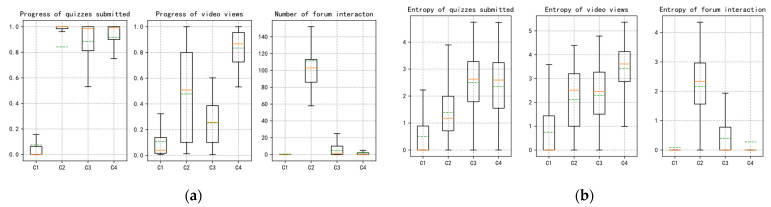
(**a**) The data distribution of the three main behavioral indicators (the left three subfigures), which represent the data distribution of quizzes submission, video viewing, and forum interaction of learners, respectively; and (**b**) the data distribution of three corresponding behavior entropies (the right three subfigures). Notes: the solid orange line represents the median and the dotted green line represents the average.

**Table 1 sensors-21-06629-t001:** The basic information of 12 courses.

Course Number	C1	C2	C3	C4	C5	C6	C7	C8	C9	C10	C11	C12
video lectures	126	154	60	15	115	62	47	78	34	31	80	113
quizzes	32	137	129	10	106	192	102	23	233	277	656	417
participants	12,693	8770	7318	7250	7157	5910	5205	5108	4242	3735	6688	2767
pass rate	44.91%	54.58%	39.90%	53.13%	16.95%	47.66%	32.14%	11.98%	38.68%	67.66%	5.92%	11.98%

**Table 2 sensors-21-06629-t002:** The statistical behavioral features of learners.

Feature Name	Feature Num	Feature Description
Progress of quizzes submitted	QP	The progress of different quizzes submitted by the learner
Score rate of quizzes submitted	QSR	The score rate of learners after submitting quizzes
Average time interval of quizzes submitted	QAT	The mean value of time interval of quizzes submitted by the learners
Max time interval of quizzes not submitted	NQT	The maximum value of time interval of quizzes submitted by the learners
Progress of video views	VP	The progress of learners watched different videos in a course
Average time of video views	VT	The average time spent by learners watching video lectures each time
Multiples of video playbacks	VM	The multiple of the video played by the learners online
Number of repeated video views	VRN	The number of video lectures repeatedly viewed by the learners online
Probability of video playback progress bar moved	VMP	The probability of the learners moved the progress bar while watching the videos
Rate of video views	VR	The proportion of time spent watching lecture videos in study time
Average time interval of video views	VAT	The mean value of time interval of video lectures viewed by the learners
Variance of time interval of video views	VTV	The variance of time interval of video lectures viewed by the learners
Time interval until first video views	VFT	The time interval from the start of the course to the first time the learners watched the video lectures
Number of forum posts	PN	The number of posts written by learners in the forums
Number of forum replies	RN	The number of posts replied by learners in the forums
Number of words posted in forum	PWN	The number of words of posted by learners in the forums
Number of words replied in forum	RWN	The number of words of replied by learners in the forums
Number of content searches	SN	The number of times a learner searches for content on the website during the course
Number of account logins	LN	The number of times the learner logged into the account during the course

**Table 3 sensors-21-06629-t003:** Pearson correlation coefficient of all course combined.

Course													
Feature	All	C1	C2	C3	C4	C5	C6	C7	C8	C9	C10	C11	C12
Number of forum posts	0.08 ***	0.10 ***	0.15 ***	0.05 **	0.02	0.23 ***	0.28 ***	0.05	0.22 ***	0.04	0.05	−0.003	0.10 ***
Number of forum replies	0.14 ***	0.15 ***	0.54 ***	0.14 ***	0.26 ***	0.62 ***	0.21 ***	0.04	0.01	0.04	0.03	0.002	0.005
Entropy of in discussion forum	0.28 ***	0.23 ***	0.39 ***	0.25 ***	0.13 ***	0.60 ***	0.16 ***	0.08 **	0.52 ***	0.08 ***	0.04	0.001	0.04
Number of account logins	0.12 ***	0.31 ***	−0.22 ***	0.13 ***	−0.21 ***	−0.12 ***	−0.12 ***	0.18 ***	−0.01	0.09 ***	−0.04	−0.07 ***	0.16 ***
Number of content searches	−0.08 ***	−0.09 ***	−0.17 ***	−0.02	−0.22 ***	−0.14 ***	−0.07 ***	−0.19 ***	−0.14 ***	−0.06 **	−0.14 ***	−0.19 ***	−0.17 ***
Progress of quizzes submitted	0.93 ***	0.98 ***	0.98 ***	0.98 ***	0.95 ***	0.89 ***	0.92 ***	0.99 ***	0.92 ***	0.77 ***	0.98 ***	0.82 ***	0.99 ***
Score rate of quizzes submitted	0.83 ***	0.89 ***	0.88 ***	0.85 ***	0.95 ***	0.69 ***	0.66 ***	0.93 ***	0.93 ***	0.61 ***	0.77 ***	0.70 ***	0.89 ***
Max time interval of quizzes not submitted	0.33 ***	0.42 ***	0.42 ***	0.14 ***	−0.10 ***	0.58 ***	0.49 ***	0.34 ***	0.74 ***	0.29 ***	0.10 ***	0.47 ***	0.61 ***
Entropy of quizzes submitted	0.68 ***	0.93 ***	0.83 ***	0.57 ***	0.28 ***	0.89 ***	0.38 ***	0.70 ***	0.81 ***	0.39 ***	0.37 ***	0.74 ***	0.93 ***
Progress of video views	0.54 ***	0.69 ***	0.49 ***	0.53 ***	0.19 ***	0.57 ***	0.36 ***	0.5 ***	0.59 ***	0.36 ***	0.12 ***	0.37 ***	0.9 ***
Average time of video views	0.35 ***	0.27 ***	0.45 ***	0.28 ***	0.25 ***	0.36 ***	0.19 ***	0.34 ***	0.52 ***	0.31 ***	0.08 **	0.34 ***	0.58 ***
Multiples of video playbacks	0.10 ***	0.31 ***	−0.10 ***	0.13 ***	−0.28 ***	−0.01	0.10 ***	0.42 ***	0.22 ***	0.16 ***	0.02	0.27 ***	0.23 ***
Number of repeated video views	−0.13 ***	−0.25 ***	−0.12 ***	−0.08 ***	−0.10 ***	0.24 ***	−0.26 ***	−0.30 ***	−0.17 ***	0.01	0.04	−0.17 ***	0.04
Probability of video playback progress bar moved	−0.12 ***	−0.08 ***	−0.11 ***	−0.03 *	−0.10 ***	0.12 ***	−0.06 **	−0.11 ***	−0.26 ***	−0.03	0.02	−0.09 ***	−0.11 ***
Rate of video views	0.62 ***	0.66 ***	0.64 ***	0.54 ***	0.43 ***	0.73 ***	0.83 ***	0.63 ***	0.81 ***	0.64 ***	0.54 ***	0.63 ***	0.91 ***
Average time interval of video views	0.002	0.02	0.02	−0.02	−0.10 ***	−0.02	0.04	0.07 **	0.15 ***	0.04	−0.07 *	0.07 ***	−0.10 ***
Variance of time interval of video views	−0.30 ***	−0.33 ***	−0.38 ***	−0.37 ***	−0.19 ***	−0.33 ***	−0.12 ***	−0.18 ***	−0.25 ***	−0.21 ***	−0.01	−0.19 ***	−0.38 ***
Time interval until first video views	−0.27 ***	−0.27 ***	−0.42 ***	0.21 ***	−0.42 ***	−0.41 ***	−0.43 ***	−0.08 **	−0.13 ***	0.02	0.004	0.04	−0.67 ***
Entropy of video views	0.64 ***	0.80 ***	0.67 ***	0.64 ***	0.39 ***	0.72 ***	0.33 ***	0.65 ***	0.81 ***	0.42 ***	0.22 ***	0.60 ***	0.92 ***
Number	34901	8413	5240	3481	2745	2866	2015	1747	1124	1821	1148	2226	2075

* *p* < 0.05, ** *p* < 0.01, *** *p* < 0.001.

**Table 4 sensors-21-06629-t004:** Cluster centroids.

Cluster Indicator	O1	O2	O3	O4
Progress of quizzes submitted (QP)	0.075	0.842	0.886	0.918
Progress of video views (VP)	0.108	0.477	0.253	0.834
Number of forum interaction (FN)	0.548	111.508	4.724	2.384

**Table 5 sensors-21-06629-t005:** The prediction results of all courses combined.

Classifier	Metric	Results of [30]	Results without Entropy	Our Model’s Results with Entropy												
				All	C1	C2	C3	C4	C5	C6	C7	C8	C9	C10	C11	C12
LR	Precision	0.818	0.864	0.892	0.943	0.980	0.911	**0.994**	0.917	0.933	**0.939**	0.921	0.928	0.993	0.756	0.938
DT		0.753	0.888	0.905	0.945	0.981	0.923	0.993	0.927	0.948	0.937	0.908	0.959	**0.995**	0.796	0.950
SVM		0.812	0.876	0.903	0.944	0.981	0.899	0.991	0.927	0.933	0.935	0.914	0.962	0.985	0.765	0.943
RF		0.824	0.917	**0.924**	**0.951**	**0.983**	**0.933**	0.993	**0.949**	**0.952**	0.936	**0.926**	**0.966**	**0.995**	**0.886**	**0.956**
LR	Recall	0.824	0.962	0.962	0.978	**0.998**	0.976	0.998	0.939	**0.981**	0.942	0.913	0.950	0.999	0.594	0.951
DT		0.752	0.951	0.959	0.978	0.995	0.981	0.998	0.930	0.978	**0.993**	**0.994**	0.956	0.999	**0.822**	0.939
SVM		0.816	0.972	0.970	**0.986**	**0.998**	**0.995**	**0.999**	**0.956**	0.933	0.966	0.957	**0.960**	**1.000**	0.590	0.945
RF		0.830	0.969	**0.972**	0.983	0.996	0.980	**0.999**	**0.956**	0.971	0.992	0.975	0.957	0.999	0.755	**0.957**
LR	F1 score	0.812	0.910	0.926	0.960	0.989	0.943	**0.996**	0.927	0.956	0.941	0.917	0.939	0.996	0.663	0.944
DT		0.752	0.920	0.931	0.961	0.988	0.951	0.995	0.928	**0.963**	**0.964**	0.948	0.957	**0.997**	0.805	0.943
SVM		0.798	0.915	0.935	0.965	**0.990**	0.944	0.995	0.941	0.957	0.950	0.935	**0.961**	0.992	0.664	0.944
RF		0.822	0.942	**0.947**	**0.967**	**0.990**	**0.956**	**0.996**	**0.952**	0.962	0.963	**0.949**	**0.961**	**0.997**	**0.813**	**0.956**
LR	AUC	0.844	0.934	0.950	0.988	**0.983**	0.960	0.989	0.988	0.977	0.967	0.972	0.962	**1.000**	0.663	0.991
DT		0.689	0.940	0.953	0.985	0.970	0.959	0.969	0.986	0.984	0.964	0.982	0.975	0.990	0.948	0.990
SVM		0.843	0.936	0.958	0.989	0.982	0.959	**0.993**	0.991	0.963	0.966	0.974	**0.981**	0.999	0.932	0.990
RF		0.851	0.967	**0.974**	**0.990**	0.981	**0.971**	0.991	**0.992**	**0.990**	**0.971**	**0.985**	**0.981**	0.982	**0.959**	**0.994**

Bold represents the maximum value.

## Data Availability

The data set from Xuetang platform used in our study was obtained from the internal share of the project National Key R&D Program of China (grant number: 2018YFB1004504).

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
