# Peer review of "MOOC Behavior Analysis and Academic Performance Prediction Based on Entropy"

_sensors, 2021, doi:10.3390/s21196629_

Round 1
Reviewer 1 Report
In general, it is a nice paper on drop-out prediction in MOOCs. There are many possible methods to solve the addressed problem. Some of these are also mentioned by authors in the introduction. It is quite convincing that the method proposed by the authors is somewhat effective. However, it is important to outline what are the main strengths of this proposed method.
Beyond the small remarks below, my main concern is related to the comparison to the previously reported results. While not arguing the effectiveness of the proposed method, it is hard for me to quantify the effectiveness compared to other (some also reported in the paper) methods that forecast students’ learning outcomes. The paper is missing a clear explanation of why the proposed method is better than the already reported ones. Moreover, a quantitative comparison is favorable.
Small remarks:
Eq. (1): please clearly state the derivation of p(xi).
Fig. 1: please consider different heatmap presentations since higher-value numbers have very low contrast.
The abstract needs proofreading with '-' marks.
Reviewer 2 Report
This paper presents the potential applications of behaviour analysis and academic performance prediction based on entropy to the
improvement of MOOC education. The authors proposed a set of quantitative behavioural indicators to examine the relationships between behaviours and academic performance and measured the prediction performance by using 4 models applied on existing MOOC datasets. Overall, the paper is well-written although there are some unclear and grammatically incorrect sentences.
However, one major concern is about the analysis of behavior entropy and more specifically how entropy was used to measure uncertainty in learner behaviour. "Temporal behavioral features" section as at the moment is difficult to understand. What makes the analysis unclear is that there is no explanation of how the probability of an indicator is calculated. The example of the entropy of video views that the learner accessed in course lectures is not very helpful. What does the probability p(x) represent in the case of video views? Therefore, I suggest the authors organise and explain further the third section.
Additionaly, the authors focus on the record of a learner’s behavior throughout the day by dividing a day into 48 thirty minutes time bins. This half-hourly basis (instead of the expected daily basis) adds to the confusion and makes the abovementioned clarification imperative.
Based on their experiments, the authors conclude that their prediction results are better than other related studies. However, some important issues and potential limitations regarding the selected features seem to be ignored. For example, counting the video views is not suitable for the situation that a learner viewed a video for seconds. Similarly, counting the viewing time length is not reasonable for the situation that a learner watched a video repeatedly. All videos have the same length?
I suggest the authors adumbrate the limitations of the proposed approach and/or clearly state the assumptions that this study subsumes.
Round 2
Reviewer 1 Report
The authors addressed the presented concerns.
Author Response
Dear reviewer,
Thank a million for allowing a resubmission of our manuscript, with an opportunity to address your comments.
We have no doubt that the comments and corrections made have helped us to produce a much improved manuscript worthy of publication.
Best regards,
Liang Zhao on behalf of myself and the co-authors.
Reviewer 2 Report
I thank the authors for their reply letter and updated paper. I do, however, have a one of comment that I feel must be addressed before the paper is accepted. In my previous comment 3, the authors answered that they divided one day into 48 half an hour time slot and cited the following paper
Reference [34]: Zhao, L.; Chen, K.; Song, J.; Zhu, X.; Sun, J. etc. Academic Performance Prediction Based on Multisource,
Multifeature Behavioral Data. IEEE Access,2021,9: 5453-5465.
However, in the abovementioned paper, the researchers seem to take into account day slots together with the time slots in order to measure student's academic behavour. Thus, I think the authors must explain how they can perform academic performance prediction without combining the sequences of day and time bins and what conclusions they can expect about student behaviour. Rearranging traditional school schedules may lead to increased academic performance but MOOCs are supposed to be accessible throughout the day. Why the time of day is so important in this analysis?
Therefore, I would ask the authors to explain better their idea and cite a more relevant work that uses only time slots to draw conclusions about student behaviour and academic performance prediction
Author Response
Dear reviewer,
Thank a million for allowing a resubmission of our manuscript, with an opportunity to address your comments. We have no doubt that the comments and corrections made have helped us to produce a much improved manuscript worthy of publication.
Best regards
Liang Zhao on behalf of myself and the co-authors
